# HOMA Index, Vitamin D Levels, Body Composition and Cardiorespiratory Fitness in Juvenile Obesity: Data from the CHILT III Programme, Cologne

**DOI:** 10.3390/ijerph19042442

**Published:** 2022-02-20

**Authors:** Annika Fraemke, Nina Ferrari, David Friesen, Fabiola Haas, Marlen Klaudius, Esther Mahabir, Lisa Schmidt, Christine Joisten

**Affiliations:** 1Department for Physical Activity in Public Health, Institute of Movement and Neurosciences, Am Sportpark Müngersdorf 6, German Sport University Cologne, 50933 Cologne, Germany; nina.ferrari@uk-koeln.de (N.F.); d.friesen@dshs-koeln.de (D.F.); f.haas@dshs-koeln.de (F.H.); m.klaudius@dshs-koeln.de (M.K.); ernaehrungschmidt@gmail.com (L.S.); c.joisten@dshs-koeln.de (C.J.); 2Cologne Center for Prevention in Childhood and Youth/Heart Center Cologne, University Hospital of Cologne, Kerpener Str. 62, 50937 Cologne, Germany; 3Comparative Medicine, Center for Molecular Medicine, University of Cologne, Faculty of Medicine and University Hospital Cologne, Robert-Koch-Str. 21, 50931 Cologne, Germany; esther.mahabir-brenner@uni-koeln.de

**Keywords:** childhood obesity, vitamin D, cardiorespiratory fitness, diabetes, HOMA index

## Abstract

Juvenile obesity is associated with insulin resistance, among other comorbidities. In the pathogenesis of insulin-resistance-related diseases, including obesity and diabetes, Vitamin D deficiency is very common. Therefore, the relationship between insulin resistance, body composition, vitamin D level, and cardiorespiratory fitness in obese children and youth were analyzed based on the Children’s Health InterventionaL Trial III project, Germany. Data on vitamin D levels and homeostatic model assessment (HOMA) indices were available from 147 participants (52.4% female; 90.5% obese; 12.3 ± 2.3 years, BMI: 30.5 ± 5.2 kg/m^2^, BMI standard deviation score (BMI-SDS): 2.52 ± 0.46). Vitamin D levels correlated negatively with the HOMA index, BMI, BMI-SDS, abdominal circumference, and body fat percentage but positively with relative cardiorespiratory fitness (*p* < 0.05 in each case). In the backward stepwise linear regression analysis, body fat (in kg; β = 0.403) and vitamin D levels (β = −0.154) explained 21.0% of the variance in the HOMA index. In summary, increased body fat and lower vitamin D levels are associated with increased HOMA indices in overweight and obese children and adolescents. In order to prevent potential negative consequences, including the development of manifest Type 2 diabetes, a healthy lifestyle with a vitamin-D-enriched diet and more time spent outdoors should be promoted.

## 1. Introduction

Globally, the number of overweight and obese children has increased. According to the World Health Organization, 5.6% of children under 5 years of age (2019) and 18% of children between 5 and 18 years (2016) were overweight [1,2]. In Germany, approximately 15% of all children and adolescents aged between 3 and 17 years have been affected [3]. Juvenile obesity has been associated with cardiovascular diseases (e.g., arterial hypertension and endothelial dysfunction), sleep apnea syndrome, psychosocial stress, and orthopedic diseases. Notably, metabolic disorders such as dyslipidemia, nonalcoholic fatty liver disease (NAFLD), insulin resistance, and even manifest Type 2 diabetes mellitus have also been associated with juvenile obesity [4,5,6,7,8].

In this context, vitamin D, or serum 25-hydroxyvitamin D-25(OH)D, also seems to play a central role. Vitamin D status in childhood has been inversely associated with cardiometabolic risk markers such as glucose, insulin, and blood pressure [9]. According to the Robert Koch Institute, in 12.5% of children aged between 1 and 17 years in Germany, vitamin D deficiency has been described [10]. Gilbert-Diamond et al. showed that vitamin-D-deficient children had an adjusted 0.1 kg/m^2^ per year greater change in BMI than did vitamin-D-sufficient children [11]. In terms of obesity, a decreased bioavailability of vitamin D is described because of its deposition in body fat compartments [12]. Additionally, visceral fat may cause metabolic abnormalities by secreting inflammatory adipokines, such as interleukin, tumor necrosis factor-α, etc., which induce insulin resistance/diabetes and vitamin D metabolic abnormalities [13]. The proposed mechanisms by which vitamin D modulates glycemic homeostasis also involve modulation of glucose-mediated synthesis and/or secretion of insulin by β-cells, increasing both hepatic and peripheral glucose uptake by direct and indirect mechanisms, and blunting inflammation [14,15]. Moreover, it is well established that normal levels of vitamin D are essential for keeping extracellular calcium concentrations and calcium influx into β-cells for insulin secretion. Vitamin D stimulates the expression of insulin receptors and, thereby, regulates insulin sensitivity. Further, it has been also reported that, in vivo, 1,25(OH)2D (calcitriol) upregulates the expression of GLUT-4 in muscle cells and promotes its translocation in animal model adipocytes [16,17]. 1,25(OH)2D additionally activates the transcription factor peroxisome proliferator-activated receptor delta (PPAR-δ), which mobilizes fatty acids in skeletal muscle and, thus, reduces free fatty acid (FFA)-induced insulin resistance [17]. Moreover, parathyroid hormone (PTH) levels increase, thus leading to an influx of calcium into the adipocytes and an increase in lipogenesis [16]. In addition to diet, a sufficient exposure to UVB light for endogenous synthesis of vitamin D, e.g., through physical activity, is necessary [18]. Another rationale for lower levels in obese people is reduced amount of exercise [19]. Thus, physical activity can be considered a proxy for sun exposure, at least in the summer months. A lower level of physical activity or lower cardiorespiratory and muscular fitness, in turn, is associated with lower insulin sensitivity and even the development of diabetes mellitus in almost all age groups [20,21]. In adults, it was also shown that, in addition to lack of exercise, obesity, and other cardiovascular risk factors, as well as sociodemographic variables, a low vitamin D level was a significant predictor for the occurrence of diabetes [22].

Corresponding data for childhood and adolescence are still inconsistent. Higher vitamin D levels, in turn, were associated with increased muscle strength in 15-year-old boys [23]. Similarly, the Healthy Lifestyle in Europe by Nutrition in Adolescence (HELENA) study showed a positive association between vitamin D and physical fitness in boys, but not in girls, in the shuttle run test of 1006 European adolescents aged 12 to 17 years [24]. Dong et al. described that a low vitamin D status was associated with obesity and lower fitness in 559 adolescents between 14 and 18 years of age in the southeastern United States of America [25]. The clinical relevance, however, has so far remained open.

Therefore, based on the Cologne Children’s Health InterventionaL Trial (CHILT) project—an outpatient training programme for obese children and adolescents—the association between vitamin D levels and the insulin resistance based on the HOMA index, as well as body composition and cardiorespiratory fitness, was analyzed. This study aimed to better assess the possible health consequences of vitamin D deficiency in obese children and adolescents and, most importantly, derive possible recommendations for action.

## 2. Materials and Methods

CHILT represents an outpatient, 11-month family-based programme for obese children and adolescents aged between 8 and 16 years, which has been running since 2003 [26]. Input data from 2008 to 2021 were integrated into this analysis. Ethical approval for this study was obtained from the German Sport University Cologne (Ethics reference number: 107/2014). The study was conducted in accordance with the ethical principles of medical research on humans (Declaration of Helsinki) and the World Medical Association. All study participants signed informed consent forms affirming their voluntary participation.

### 2.1. Study Population

From 536 children and adolescents, 147 were integrated into the study (see Figure 1). Participants were included for whom data were available on the following parameters: sex, age, height, weight, abdominal circumference, BMI, BMI standard deviation score (BMI-SDS), body fat content, relative and absolute watt, levels of vitamin D, fasting blood glucose, insulin, and the HOMA index (2008 to 2021). Girls formed 52.4% of the participants, the mean age was 12.3 ± 2.3 years, the BMI was 30.5 ± 5.2 kg/m^2^, and the BMI-SDS was 2.52 ± 0.5.

### 2.2. Anthropometric Data

Height was measured barefoot in centimeters using a stadiometer, and weight was measured in kilograms using calibrated scales [27]. The BMI was calculated using the formula body weight (kg)/(body height (m))^2^ and classified according to the percentiles by Kromeyer-Hauschild et al. [28]. A BMI above the 90th percentile was classified as overweight and a BMI above the 97th percentile as obese [29]. The BMI-SDS was calculated using the least-mean-squares (LMS) method for non-normally distributed characteristics [29]. L describes the Box–Cox transformation, M the median, and S the coefficient of variation. The age-specific parameters L(t), M(t), and S(t) are the starting point for calculating the percentiles according to the following formula (1):(1)$${\mathrm{SDS}}_{\mathrm{LMS}}=\frac{\left(\mathrm{BMI}/M\left[t\right]L\left[t\right]\;–\;1\right)\;}{\left(L\left[t\right]\times \;S\left[t\right]\right)\;}$$

Waist circumference was measured in centimeters with the children standing upright using a standard tape measure. The measurements were taken midway between the *spina iliaca* anterior superior and the lowest rib.

Skinfold thickness was measured with a body fat caliper (Harpender Skinfold Caliper HSK-BI, British Indicators, West Sussex, UK) to the nearest 0.2 mm in triplicate in the triceps and subscapular skinfolds according to a standardized protocol [30], reporting the mean of the three results.

On this basis, the body fat percentage was used according to the formulas of Slaughter and Rodríguez et al. [31,32]. Muscle mass was determined by a four-point bioelectrical impedance analysis (BIA; Nutriguard-MS, Data Input GmbH, Pöcking, Germany). The frequency of the measurement was 50 kHz [33]. The following values were determined from the measurement: resistance®, reactance (Xc), test buzzer (Σ), total resistance (Rtot.), and phase angle (φ). With the help of the programme NutriPlus (NutriPlus, Data Input GmbH, Pöcking, Germany), muscle mass (kg) was calculated from these values and additionally given as a percentage [34].

### 2.3. Laboratory Parameters

A venous blood sample (7.5 mL serum tube, S-Monovette, Sarstedt, Nümbrecht, Germany) was obtained in a fasting state (>12 h fasting). The samples were centrifuged at 4000 rpm for 10 min at 4 °C in a Hettich MR20 centrifuge (Tuttlingen, Germany). Then, the serum was removed and placed in a new tube for storage at −80 °C until evaluation. Fasting blood glucose levels were measured directly after blood collection. Insulin levels were determined using human insulin standards (Elecsys Insulin) from Roche Diagnostics, Mannheim (supplement slip) [35].

The HOMA index was calculated according to [36], as shown in Equation (2):(2)$$\mathrm{HOMA}=\frac{\left(\mathrm{insulin}\;\left[\mathrm{mU}/L\right]\times \;\mathrm{glucose}\;\left[\mathrm{mmol}/L\right]\right)}{22.5}$$

No uniform cutoffs currently exist for the classification of the HOMA index. In a systematic review or meta-analysis, Arellano-Ruiz et al. reported the cutoff as being between 2.30 and 3.59. In the context of this analysis, a value > 3.0 was, therefore, assumed to be an approximation for insulin resistance [37].

The 25(OH)D was measured using a competitive binding assay (Elecsys Vitamin D total II (Roche Diagnostics)) and analyzed using Cobas E801 (Roche Diagnostics). According to the Robert Koch Institute, an optimal 25(OH)D level between the ages of 1 and 17 years is >20 ng/mL, a suboptimal level is 12–20 ng/mL, and a deficiency is <12 ng/mL [11].

### 2.4. Determination of Cardiorespiratory Fitness

Cardiorespiratory fitness (in watts) was determined by bicycle ergometry (Ergometrics er900, Ergoline, Bitz, Germany). The children started at 25 watts; workload was increased by 25 watts every 2 min until the maximum subjective workload was reached. Relative performance was defined as the maximum watt in relation to body weight, in watts/kg (W/kg) [26].

Children with acute diseases such as febrile infections, asthma attacks, or metabolic diseases were excluded from ergometric testing. Other contraindications were cardiomyopathies, certain vascular anomalies, and heart failure [38].

### 2.5. Statistical Analysis

The data were analyzed using the IBM programme SPSS Statistics version 28.0 (IBM Corp., Armonk, NY, USA), and descriptive statistics were presented as means and standard deviations. Means of continuous parameters were compared using a *t*-test. Categorical parameters were tested using a chi-square test (X^2^). Backwards multiple linear regression analysis was used to examine the factors potentially influencing the HOMA index such as age (in y), gender (male = 1, female = 2), BMI (in kg/m^2^), BMI-SDS, absolute and relative watt (w/kg body weight), absolute (in kg) and relative body fat percentage (in %), absolute (in kg) and relative muscle mass (in %), and vitamin D levels (in ng/mL). Nonsignificant factors were excluded during stepwise regression. The significance level was defined as a *p*-value < 0.05.

## 3. Results

### 3.1. Anthropometric Data and Laboratory Parameters

In total, 133 participants (90.5%) were obese and 11 (7.5%) were overweight. No BMI classification was calculated for three children due to missing individual data. The mean BMI-SDS was 2.52 ± 0.46. Boys were significantly heavier (*p* = 0.040) and had a higher waist circumference (*p* = 0.001). There were no other gender-related differences among the physical variables (see Table 1). Laboratory parameters were not significantly influenced by gender (see Table 1).

### 3.2. Cardiorespiratory Fitness

The absolute and relative fitness did not differ between boys and girls (see Table 1). Differences in relation to a low or sufficient vitamin D level or the presence of insulin resistance can be seen in Table 2. Participants with lower vitamin D levels had significantly lower absolute and relative cardiorespiratory fitness (*p* < 0.05). Participants with an elevated HOMA index were heavier and taller and had a higher waist circumference or absolute fat mass.

### 3.3. Regression Analyses

Vitamin D levels correlated inversely with weight, waist circumference, BMI or BMI-SDS, body fat mass, insulin levels, and the HOMA index. The HOMA index levels correlated with age, weight, BMI-SDS, body fat mass, cardiorespiratory fitness, and vitamin D levels (with *p* < 0.05 for each parameter; see data Supplement Appendix A). Linear regression was used to test the influence on the HOMA index (see Table 3). The baseline and final models are shown. An increased body fat percentage (in kg; β = 0.403; *p* < 0.001) and a lower vitamin D level (β = −0.154; *p* = 0.058) explained 21.0% (corr. R^2^) of the variance.

## 4. Discussion

To our knowledge, this is one of the first studies to analyze the relationship between vitamin D, fitness, and insulin resistance. In our analysis, lower vitamin D levels were associated with a significantly higher HOMA index as well as substantially lower cardiorespiratory fitness. Participants with an elevated HOMA index also had a higher BMI or BMI-SDS and body fat percentage. Twenty-one percent (21%) of the variance of the HOMA index was explained by an increased body fat percentage and a tendency to a lower vitamin D level.

Thus, our results confirmed the known association between juvenile obesity, low vitamin D levels, and insulin resistance. The role of physical fitness is not quite as clear; data in the literature is also rather sparse and inconsistent. In most cases, such correlations were presented for children who were active in sports and confirmed the natural influence of the seasons [39,40]. In general, overweight children tend to perform less well and are more likely to be inactive [41]. Chee et al. showed, in 243 Malaysian preschool-aged children, that body surface area exposed to sunlight, including outdoor exercise, solar index, and fat mass, were significant predictors of 25(OH)D concentration in this population [42]. In the Danish Optimal well-being, development, and health for Danish children through a healthy New Nordic Diet (OPUS) school meal study, moderate physical activity was associated with higher levels of vitamin D [43]. In this study, physical activity was objectively measured via accelerometers. In terms of the association between measured cardiorespiratory fitness and vitamin D levels in obese children, only two studies have been conducted so far.

In the HELENA study mentioned earlier, low 25(OH)D levels were associated with lower endurance performance in the shuttle run test and a higher BMI in boys [24]. In a recent study, no association was evident between the levels of vitamin D and relative or maximal oxygen uptake in 57 prepubertal overweight children [44]. However, the authors pointed out that their data should be interpreted with caution due to the small and heterogeneous group. In our study, the correlations were also not entirely clear: children with insufficient vitamin D levels also had significantly lower absolute and relative cardiorespiratory fitness. To what extent this is clinically relevant, however, can only be speculated, because in the final model of the stepwise regression analysis, only the body fat percentage and the reduced vitamin D level tended to remain as influencing variables on the HOMA index. At present, it can only be speculated to what extent the determination of muscle strength would have led to different results due to its association with vitamin D levels [45]. In general, however, the benefits of exercise and, especially, physical fitness are known, even in childhood and adolescence. Medrano et al. showed that fit or active children had lower visceral, subcutaneous, and intramuscular adipose tissue levels (all *p* < 0.03) than their untrained or inactive counterparts [46]. In addition, a higher fitness was associated with a lower HOMA-IR [21].

### Strength and Limitations

One strength is the measurement of cardiorespiratory fitness according to standardized and objective methods in a large cohort. However, the determination of vitamin D levels was only included later in the programme, so that data from the complete CHILT collective were unavailable. The main limitation of this analysis, however, was the cross-sectional design, which does not allow for drawing conclusions on a causal relationship between the observed variables. In addition, several factors relevant to obesity and vitamin D levels, such as dietary habits, were not considered due to incomplete data. Although we recorded these variables, the data on diet, in particular, are subject to social desirability bias. The decrease in vitamin D levels consistently occurred in the sunnier summer months, which may also have influenced the level of the values. In addition, self-reports on physical activity and media consumption were not considered in this analysis, as the study focused on using objective methods. In further studies, muscle strength and sedentary behavior should also be analyzed by accelerometry to assess the clinical significance of low vitamin D levels. This relationship should also be tested in longitudinal studies.

## 5. Conclusions

In this study, increased body fat and low vitamin D levels were associated with an increased HOMA index in overweight or obese children and adolescents. In addition, lower cardiorespiratory performance was related to lower vitamin D levels. To ensure optimal vitamin D levels in this population and to prevent possible negative consequences including the development of manifest Type 2 diabetes, a healthy, active lifestyle with a vitamin-D-rich diet, sufficient sunlight exposure, and more time outdoors should be promoted.

## Figures and Tables

**Figure 1 ijerph-19-02442-f001:**
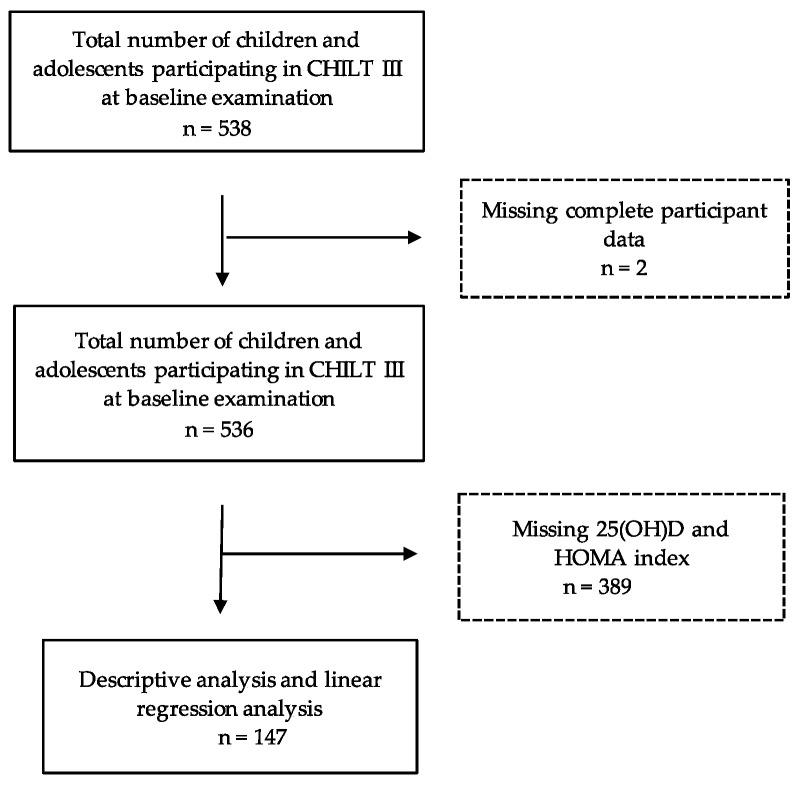
Study population.

**Table 1 ijerph-19-02442-t001:** Percentiles, physical variables, and laboratory parameters in girls and boys.

Variable		Total	Boys	Girls	*p*-Value
Percentile	Obese	133 (90.5%)	62 (88%)	71 (96%)	0.096 ^†^
	Overweight	11 (7.5%)	8 (12%)	3 (4%)	
Physical Variables	Age (years)	12.3 ± 2.3 (*n* = 147)	12.5 ± 2.1 (*n*= 70)	12.0 ± 2.4 (*n* = 77)	0.191 ^‡^
	Height (m)	1.57 ± 0.1 (*n* = 144)	1.59 ± 0.1 (*n* = 70)	1.55 ± 0.1 (*n* = 74)	0.058 ^‡^
	Weight (kg)	76.4 ± 19.9 (*n* = 144)	79.9 ± 23.1 (70)	73.1 ± 15.8 (*n* = 74)	0.043 ^‡^
	Waist circumference	95.2 ± 13.6 (*n* = 136)	99.0 ± 16.0 (*n* = 68)	91.3 ± 9.2 (*n* = 68)	<0.001 ^‡^
	BMI (kg/m^2^)	30.5 ± 5.2 (*n* = 144)	31.1 ± 6.2 (*n* = 70)	29.9 ± 3.9 (*n* = 74)	0.188 ^‡^
	BMI SDS	2.52 ± 0.5 (*n* = 144)	2.49 ± 0.5 (*n* = 70)	2.55 ± 0.4 (*n* = 74)	0.478 ^‡^
	Body fat (kg)	19.8 ± 6.3 (*n* = 137)	20.8 ± 7.4 (*n* = 68)	18.8 ± 4.9 (*n* = 69)	0.065 ^‡^
	Body fat (%)	25.4 ± 2.2 (*n* = 137)	25.5 ± 2.3 (*n* = 68)	25.3 ± 2.1 (*n* = 69)	0.609 ^‡^
	Muscle mass (kg)	23.1 ± 7.2 (*n* = 73)	24.0 ± 8.7 (*n* = 37)	22.3 ± 5.1 (*n* = 36)	0.324 ^‡^
	Muscle mass (%)	30.8 ± 4.2 (*n* = 73)	30.4 ± 5.1 (*n* = 37)	31.3 ± 2.9 (*n* = 36)	0.361 ^‡^
Fitness parameter	Absolute cardiorespiratory Fitness (W)	120.0 ± 33.9 (*n* = 135)	124.5 ± 38.4 (*n* = 68)	115.4 ± 28.2 (*n* = 67)	0.188 ^‡^
	Relative cardiorespiratory Fitness (W/kg)	1.6 ± 0.4 (*n* = 135)	1.6 ± 0.4 (*n* = 68)	1.6 ± 0.4 (*n* = 67)	0.404 ^‡^
Laboratory parameters	25(OH)D (ng/mL)	23.29 ± 9.8 (*n* = 147)	24.88 ± 10.9 (*n* = 70)	21.85 ± 8.6 (*n* = 77)	0.062 ^‡^
	Blood glucose (mg/dL)	90.76 ± 7.9 (*n* = 147)	91.72 ± 7.8 (*n* = 70)	89.89 ± 7.9 (*n*= 77)	158 ^‡^
	Insulin (µU/mL)	22.27 ± 11.8 (*n* = 147)	23.00 ± 12.7 (*n* = 70)	21.61 ± 10.9 (*n*= 77)	0.478 ^‡^
	HOMA index	5.01 ± 2.7 (*n* = 147)	5.20 ± 2.8 (*n* = 70)	4.84 ± 2.5 (*n* = 77)	0.419 ^‡^

The data are presented as the mean ± SD; BMI = body mass index; SDS = standard deviation score; HOMA = homeostasis model assessment; calculated with the ^†^ chi-square test and ^‡^ *t*-test.

**Table 2 ijerph-19-02442-t002:** Selected physical and laboratory parameters considering the vitamin D level and the HOMA index, respectively.

Variable		Vitamin D (ng/mL)			HOMA Index		
		Lowered Vitamin D Level	Sufficient Vitamin D Level	*p*-Value	Insulin Resistance	Absence of Insulin Resistance	*p*-Value
Physical Variables	Age (years)	12.9 ± 2.2	12.2 ± 2.3	0.276 ^‡^	12.4 ± 2.2	11.6 ± 2.4	0.078 ^‡^
	Height (m)	1.56 ± 0.1	1.57 ± 0.1	0.747 ^‡^	1.56 ± 0.1	1.52 ± 0.1	0.004 ^‡^
	Weight (kg)	78.3 ± 21.8	76.2 ± 19.8	0.701 ^‡^	79.0 ± 20.3	66.0 ± 14.7	0.001 ^‡^
	Waist circumference (cm)	100.3 ± 15.4	94.7 ± 13.4	0.195 ^‡^	97.2 ± 13.8	86.9 ± 8.8	<0.001 ^‡^
	BMI (kg/m^2^)	31.6 ± 6.4	30.4 ± 5.0	0.395 ^‡^	31.0 ± 5.4	28.3 ± 3.6	0.010 ^‡^
	BMI SDS	2.54 ± 0.6	2.52 ± 0.4	0.861 ^‡^	2.55 ± 0.5	2.39 ± 0.4	0.098 ^‡^
	Body fat (kg); Caliper	20.8 ± 7.3	19.7 ± 6.2	0.564 ^‡^	20.4 ± 6.6	17.0 ± 4.2	0.002 ^‡^
	Body fat (%); Caliper	25.5 ± 2.6	25.4 ± 2.1	0.870 ^‡^	25.6 ± 2.2	24.9 ± 1.9	0.161 ^‡^
	Muscle mass (kg)	21.0 ± 6.3	23.3 ± 7.3	0.225 ^‡^	18.7 ± 3.4	23.9 ± 7.4	0.026 ^‡^
	Muscle mass (%)	29.1 ± 2.4	31.0 ± 4.3	0.148 ^‡^	29.9 ± 3.9	31.0 ± 4.2	0.422 ^‡^
Fitness parameter	Absolute cardiorespiratory fitness (W)	100.0 ± 16.7	121.6 ± 34.5	0.003 ^‡^	121.7 ± 34.5	112.5 ± 30.7	0.460 ^‡^
	Relative cardiorespiratory fitness (W/kg)	1.3 ± 0.3	1.6 ± 0.4	0.004 ^‡^	1.6 ± 0.4	1.7 ± 0.3	0.529 ^‡^
Laboratory parameters	Vitamin D (ng/mL)				22.6 ± 9.3	25.97 ± 11.5	0.095 ^‡^
	Blood glucose (mg/dL)	5.13 ± 0.5	5.02 ± 0.43	0.148 ^‡^			
	Insulin (µU/mL)	27.00 ± 8.8	21.78 ± 8.6	0.115 ^‡^			
	HOMA index	6.19 ± 2.2	4.88 ± 2.7	0.082 ^‡^			

The data are presented as the mean ± SD; BMI = body mass index; SDS = standard deviation score; HOMA = homeostasis model assessment; *p*-values calculated with the ^‡^ *t*-test.

**Table 3 ijerph-19-02442-t003:** Baseline and final models from backward stepwise multivariable linear regression analysis.

Model		Beta (s.e.)	*p*-Value	R^2^ (corr.)
Baseline	Age (years)	0.047 (0.226)	0.799	0.205
	Sex	−0.033 (0.476)	0.717	
	BMI (kg/m^2^)	−0.789 (0.203)	0.047	
	BMI-SDS	0.325 (1.564)	0.235	
	Absolute body fat (kg); Caliper	1.315 (0.172)	0.002	
	Relative body fat (%); Caliper	−0.296 (0.197)	0.067	
	Absolute physical fitness (W)	−0.282 (0.019)	0.254	
	Relative physical fitness (W/kg)	0.188 (1.261)	0.323	
	Vitamin D (ng/mL)	−0.149 (0.023)	0.079	
Final	Vitamin D (ng/mL)	−0.154 (0.022)	0.058	0.210
	Absolute body fat (kg); Caliper	0.403 (0.033)	<0.001	

BMI = body mass index; SDS = standard deviation score; s.e.: standard error.

## Data Availability

The data used and analyzed during the current study involve sensitive patient information and indirect identifiers. As a result, the datasets are available from the corresponding author only upon reasonable request.

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
