# Peer review of "HOMA Index, Vitamin D Levels, Body Composition and Cardiorespiratory Fitness in Juvenile Obesity: Data from the CHILT III Programme, Cologne"

_ijerph, 2022, doi:10.3390/ijerph19042442_

Round 1

Reviewer 1 Report

The authors tried to assess the possible health consequences of vitamin D deficiency in obese children and adolescents and, most importantly, derive possible recommendations for action. I believe the topic is very important in terms of contribution of cardiorespiratory fitness to vitamin D deficiency in obese children and adolescents. After careful review, I would concluded that this manuscript to meet the IJERPH, if just a few improved such as following.

  1. There is a lack of possible mechanisms to accept conclusions of this study.

:  In the discussion section, please describe the possible mechanism that might support the results of this study.

Reviewer 2 Report

Research on HOMA Index, Vitamin D Levels, Body Composition and Cardiorespiratory Fitness in Juvenile Obesity has some practical implications. The design of the article is reasonable and the level is clear. The content of the Statistical analysis section of this study should be described in further detail. In addition, the analysis in the Discussion section should be further added.

Reviewer 3 Report

General Comments

This manuscript has the potential to be interesting. The Authors are requested to better explore the literature to improve the rationale of the study more in detail. Moreover, they are invited to discuss the final regression model results more in deep taking into account the explained variance along with the significance.

I suggest more specific comments:

Specific Comments

Introduction

Lines 67-89: the authors are recommended to better explain why vitamin D should be related/associated with physical activity. Moreover, the role of physical domain is less developed and undermined, but it of extremely importance when dealing with overweight and obese conditions. The following refs would help to reinforce the rationale of the study, which is weak at this stage.

1) The role of vitamin D, obesity and physical exercise in regulation of glycemia in Type 2 Diabetes Mellitus patients

doi: 10.1016/j.dsx.2016.06.007

2) Associations of physical activity with vitamin D status depends on obesity status in old adults

https://doi.org/10.1016/j.clnesp.2020.06.009

3) Actual and Perceived Motor Competence in Relation to Body Mass Index in Primary School-Aged Children: A Systematic Review

https://doi.org/10.3390/su13179994

4) Physical Illiteracy and Obesity Barrier: How Physical Education Can Overpass Potential Adverse Effects? A Narrative Review.

https://doi.org/10.3390/su14010419

Methods

Line 118: clarify the terms M(t) and S(t)

line 127: “and” instead of “or”?...please revise the citation style.

Line 158: on spiroergometry? Why? Please justify

Lines 169-170: which regression technique? Backward stepwise? Please provide it.

Discussion

Please discuss also the non-significant results of Vitamin D coming from the final regression model. Moreover, although significant for Absolute body fat, the explained variance is definitely low. I believe that this should be acknowledge within the discussion section.

Line 217: I am not surprised. Physical performance is the results of multidimensional determinants that all concur to its outcomes. It would help the reader to clarify why the authors sought to examine the association between vitamin D and physical performance.

Lines 232: Vitamin D was not sign.

Line 234: Strength is crucial, but I would not speculate in such way. Please see the literature about the crucial role of strength in pediatric obese. The papers suggested above may be right for this purpose.

Round 2

Reviewer 3 Report

I have no other comments to provide.